# Loss of *NARS1* impairs progenitor proliferation in cortical brain organoids and leads to microcephaly

Lu Wang[1,2], Zhen Li[1,2], David Sievert[1,2,10], Desirée E. C. Smith[3,10], Marisa I. Mendes[3,10], Dillon Y. Chen[2,4,5], Valentina Stanley[1,2], Shereen Ghosh[1,2], Yulu Wang[6], Majdi Kara[7], Ayca Dilruba Aslanger[8], Rasim O. Rosti[1,2], Henry Houlden [9], Gajja S. Salomons[3] & Joseph G. Gleeson [1,2,4,5]✉

Asparaginyl-tRNA synthetase1 (NARS1) is a member of the ubiquitously expressed cytoplasmic Class IIa family of tRNA synthetases required for protein translation. Here, we identify biallelic missense and frameshift mutations in *NARS1* in seven patients from three unrelated families with microcephaly and neurodevelopmental delay. Patient cells show reduced NARS1 protein, impaired NARS1 activity and impaired global protein synthesis. Cortical brain organoid modeling shows reduced proliferation of radial glial cells (RGCs), leading to smaller organoids characteristic of microcephaly. Single-cell analysis reveals altered constituents of both astrocytic and RGC lineages, suggesting a requirement for *NARS1* in RGC proliferation. Our findings demonstrate that *NARS1* is required to meet protein synthetic needs and to support RGC proliferation in human brain development.

[1] Department of Neurosciences, Howard Hughes Medical Institute, University of California San Diego, La Jolla, CA 92093, USA. [2] Rady Children's Institute for Genomic Medicine, Rady Children's Hospital, San Diego, CA 92123, USA. [3] Metabolic Unit, Department of Clinical Chemistry, Amsterdam UMC, Vrije Universiteit Amsterdam, Amsterdam Neuroscience, Amsterdam Gastroenterology & Metabolism, Amsterdam, Netherlands. [4] Department of Pediatrics, University of California San Diego, La Jolla, CA 92093, USA. [5] Division of Child Neurology, Rady Children's Hospital, San Diego, CA 92123, USA. [6] Laboratory of Biomanufacturing and Food Engineering, Institute of Food Science and Technology, Chinese Academy of Agricultural Sciences, Beijing 100193, PR China. [7] University of Tripoli, Tripoli Children's Hospital, Tripoli, Libya. [8] Department of Medical Genetics, Koç University Hospital, Istanbul 34010, Turkey. [9] Department of Neuromuscular Disorders, UCL Institute of Neurology, Queen Square, London WC1N 3BG, UK. [10] These authors contributed equally: David Sievert, Desirée E. C. Smith, Marisa I. Mendes. ✉email: jogleeson@health.ucsd.edu

Aminoacyl-tRNA synthetases (ARSs) are a ubiquitously expressed group of highly specialized enzymes mediating the charging of amino acids onto cognate transfer RNAs (tRNAs) in both cytoplasm and mitochondria, essential for protein translation[1,2]. Among the ARS proteins encoded by 37 ARS genes, responsible for the 20 typical amino acids, 17 function in the cytoplasm, 17 function in mitochondria, and 3 function in both cellular compartments[3–6]. Asparaginyl-tRNA synthetase1 (NARS1) belongs to the class IIa family, based upon a 7 beta-strand protein structure. There are two NARS genes: NARS1 functions in the cytoplasm while NARS2 functions in mitochondria, solely responsible for asparagine tRNA charging in these locations.

While canonical ARS function is conserved across all 3 branches of life, several mammalian ARSs acquired additional domains with unique structural characteristics that account for non-canonical functions. These domains are implicated in signaling transduction, cell migration, tumorigenesis, cell proliferation, and cell death[1,7,8]. NARS1 similarly contains an 77 amino acid of Unique N-terminal Extension (UNE-N) domain of unknown function[9,10].

Mutations in ARS genes have been implicated in a broad range of human disorders, including neurological, autoimmune, and cancer[1,2]. Both heterozygous and homozygous disease-causing variants in several ARS genes have been reported[11]. Biallelic damaging variants in ARS2 genes, encoding mitochondrial-localized enzymes, tend to cause mitochondrial encephalopathies, whereas biallelic damaging variants in ARS1 genes, encoding cytoplasm-localized enzymes, tend to cause epileptic encephalopathies or other systemic conditions[3,4,12–15]. Interestingly, certain variants in ARS genes show peripheral neuropathy (i.e., Charcot–Marie–Tooth syndrome) with dominant inheritance, at least partially explained by toxic gain-of-function effects of mutant proteins binding to neuropilin-2[16–22].

Primary microcephaly is a neurodevelopmental disorder in which brain volume is markedly reduced, directly reflecting a smaller cerebral cortex[23], with occipito-frontal head circumference (OFC) >2–3 standard deviations (SD) below the mean[24–26]. The causes of microcephaly are numerous, including prenatal exposure to toxins such as alcohol, in utero infections such as Zika virus, and metabolic/genetic factors[23,24,27,28]. Mutations in genes involved in centriolar biogenesis/assembly (MCPH), microtubule regulation (TUB genes), DNA damage (PNKP), signaling pathways (ALFY/ASPM), transcription and metabolism are linked to microcephaly[29–35]. Recently, QARS, KARS, VARS, and CARS ARS family genes were implicated in microcephaly, but mammalian model systems and mechanisms remain unexplored[3,4,11,36,37].

In this study, we identify seven affected individuals with neurodevelopmental defects and microcephaly, from three unrelated families, carrying biallelic damaging variants in NARS1. Patient-derived cell lines show loss of NARS1 protein function, including reduced protein expression and tRNA synthetic activity. Patient iPSC-derived cortical brain organoids (COs) are dramatically smaller than controls, correlating with microcephaly seen in patients. Single-cell RNA-seq (sc-RNA-seq) and histological assessment show transcriptional changes indicating cell proliferation and cell-cycle defects affecting RGCs. Together, these results suggest that NARS1 plays a critical role in regulating proliferation of RGCs, leading to microcephaly when deficient.

## Results

### Identification of NARS1 mutations in families with microcephaly.
Whole-exome sequencing (WES) in our cohort of over 5000 individuals with neurodevelopmental disorders led to the identification of genetic variants as candidates for disease[38]. The GATK workflow was used to identify variants with high likelihood for pathogenicity[39]. Rare, potentially deleterious variants were prioritized against in-house and public exome datasets, cumulatively numbering over 100,000 individuals[40]. Seven patients from three unrelated families with likely damaging biallelic variants in NARS1 were identified. Family MIC-1433 from Libya had three affected individuals and Family MIC-91 from India had two affected individuals both from consanguineous marriages. Family MIC-2116, from Turkey had two affected individuals, and parents denied consanguinity (Fig. 1a, Table 1, Supplementary Note 1). All affected individuals showed microcephaly, developmental delay, and intellectual disability, with progressive loss of neurological function.

In the families with parental consanguinity, homozygous variants were identified, whereas in the family without parental consanguinity, a compound heterozygous mutation was identified. A homozygous missense variant (c.50C > T, p.T17M) was found in MIC-1433. A homozygous missense variant (c.1633C > T, p.R545C) was found in MIC-91, and both falling within a block of homozygosity (Supplementary Fig. 1a, b). Compound heterozygous variants (c.[203dupA]; [1067A > C]; p.[M69Dfs*4]; [D356A]) were found in MIC-2116 (Fig. 1a). Brain MRI from the affected in 1433-III-3 showed severe microcephaly, diffuse cerebral and cerebellar atrophy, with both gray and white matter volume loss, and ventriculomegaly (Fig. 1b). The MRI from the affected in 91-III-1 and 2116-III-1, obtained much earlier in the course of disease, showed mild cortical and cerebellar atrophy, and enlarged lateral ventricular size (Fig. 1c, d). MRI reports for the other affected individuals (the scans were lost or destroyed) were consistent with these findings. We conclude that biallelic loss of NARS1 leads to progressive loss of cortical volume, consistent with the neurodegenerative clinical course.

Candidate variants were validated by Sanger sequencing in all available blood samples and reprogrammed iPS cells from MIC-1433 and MIC-2116 and segregated according to a recessive mode of inheritance (Fig. 1e). All variants fell within the NARS1 coding regions (Fig. 1f). Two variants (including the frameshifting variant in MIC-2116) were located in the UNE-N domain, which comprises part of the "Appended Domain" that are specific ARS enzymes (Fig. 1g, h); while the other two were located in the catalytic domain (CD) (Fig. 1g, i, j)[7,41]. Sequence alignment of NARS1 from vertebrates showed that the residues mutated in patients were conserved across all mammals and most vertebrates (Fig. 1k). Together, these results implicate NARS1 biallelic variants as disease-causing in these individuals.

### Patient-derived cells display loss of NARS protein function.
GTEX showed NARS1 was expressed in all human tissues including brain (Supplementary Fig. 1c). To determine the effects of mutations on NARS1 protein, whole-cell lysates were generated from patient-derived fibroblasts (see "Methods"). Western blot analysis showed NARS1 level in patient cells reduced by about half (Fig. 2a, b, Supplementary Fig. 2a, b). Overexpression of NARS1 harboring patient mutations confirmed that both missense variants rendered NARS1 unstable (Fig. 2c). We next measured NARS1 tRNA charging activity in the cytoplasm of fibroblasts, using a stable isotope pulse labeling protocol (see methods). Both patient cell lines had activity reduced by about half (Fig. 2d). As controls, we additionally assessed charging activity of three other ARSs: RARS, KARS, and TARS (for arginine, lysine, and threonine) and found patient cells had normal activity (Fig. 2e). We conclude that NARS1 is expressed in the brain, that missense variants impact protein stability, and that patient cells display reduced NARS1 protein levels and activity.

NARS1 is presumed to be solely responsible for the generation of charged cytoplasmic asparagine tRNAs in every cell.

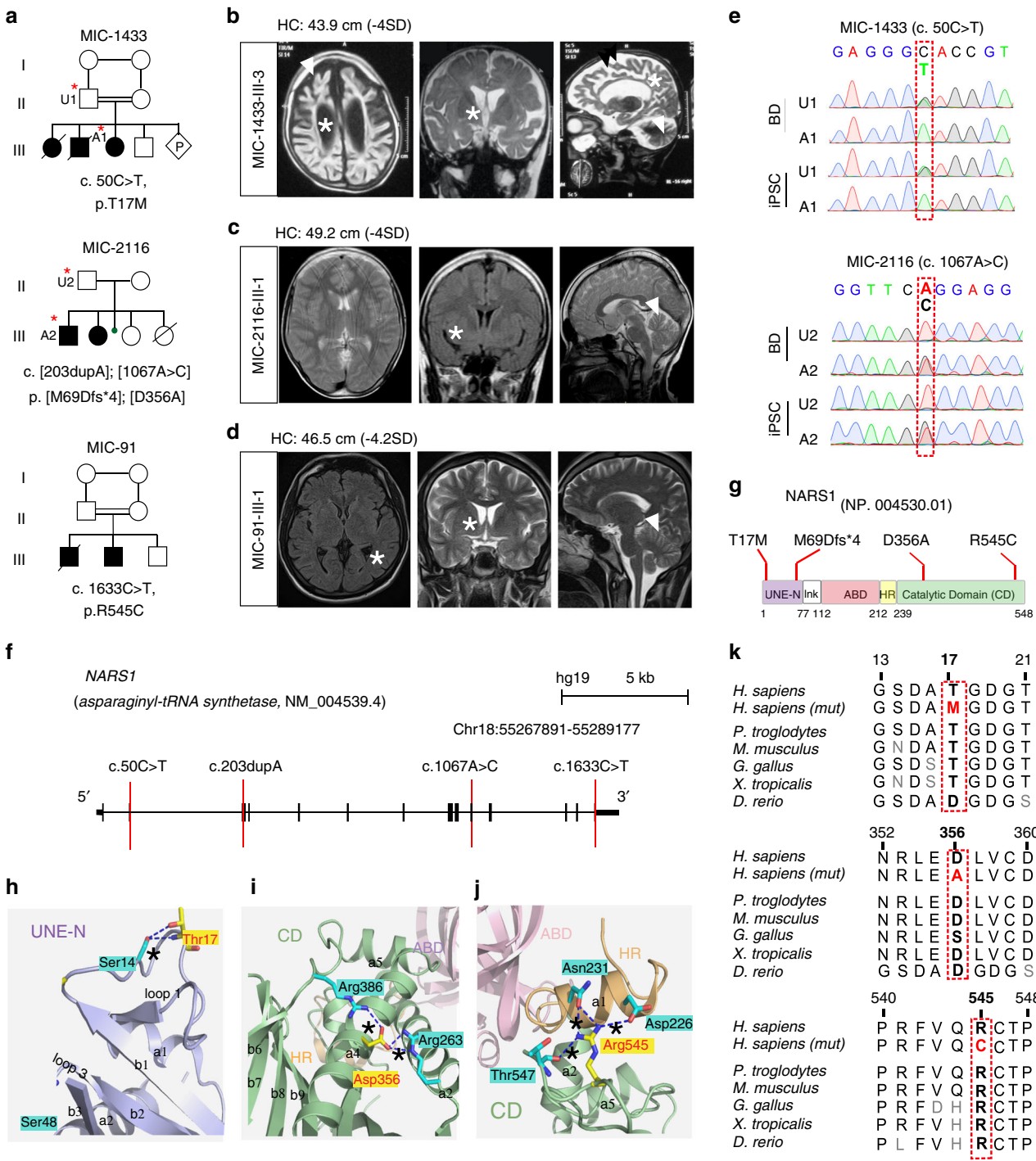

**Fig. 1 Identification of *NARS1* mutations in families with microcephaly. a** Pedigrees MIC-1433, MIC-2116, and MIC-91. Double line: consanguinity. Square: male; Circle: female; filled: affected; hash line: deceased; diamond: pregnant. U1, A1, U2, A2 (red asterisks) represent patients cell lines used in the following figures. **b–d** MRIs for affected individuals showing ventriculomegaly (asterisks) and cerebral and cerebellar atrophy (arrows). Arranged in axial, coronal, and sagittal planes. **e** Chromatogram of mutations. **f** N*ARS1* genomic organization. Red lines: mutation position according to NM_004539.4. **g** NARS1 domains: N-terminal extension (UNE-N, aa1-77), anticodon-binding (ABD, aa112-212), hinge (HR, aa213-239) and catalytic (aa240-548), (h-j). lnk: linker. NARS1 Thr17 (**h**), Asp356 (**i**) and Arg545 (**j**) highlighted. Asterisk (*) indicates hydrogen bond. **k** Conservation of mutated residue across evolution.

Puromycin is incorporated into newly synthesized proteins as a mimic tRNA, so was used to pulse label cells for 1 h to asses global protein synthesis (see methods)[42]. In cells from healthy parents, we found robust labeling, but lines from one affected individual from both families showed dramatic reduction of labeling (Fig. 2f, Supplementary Fig. 2c), suggesting reduced total protein synthesis.

**NARS1 cortical brain organoids generated from patient iPSCs model microcephaly.** In order to understand how NARS1 loss leads to microcephaly, we first generated neuronal progenitor cells (NPCs) in 2D cultures, generated from patient-derived iPSCs from two patients and two controls (one from each of MIC-1433 and MIC-2116). Using standard methods, we validated iPSCs integrity, excluding detectable chromosomal aberrations

**Table 1 Clinical information for seven affected from three families.**

| Proband | 1433-3-1 | 1433-3-2 | 1433-3-3 | 2116-3-1 | 2116-3-2 | 91-3-1 | 91-3-2 |
|---|---|---|---|---|---|---|---|
| Current age | 10 yrs (deceased) | 8 yrs (deceased) | 8 yrs | 22 yrs | 15 yrs | 33 yrs (deceased) | 17 yrs |
| Gender | Female | Male | Female | Male | Female | Male | Male |
| Parent Country of origin | Libya | | | Turkey | | India | |
| Consanguinity | + | | | − | | + | |
| **Variant** | | | | | | | |
| Zygosity | Homozygous | | | Compound Heterozygous | | Homozygous | |
| Genomic (hg19) | chr18:g.55287844G>A | | | chr18:g.55283097C>CT; chr18:g.55273918T>G | | chr18:g.55273352G>A | |
| cDNA | c.50C>T | | | c.[203dupA]; [1067A>C] | | c.1633C>T | |
| Protein | p.T17M | | | p.[M69Dfs*4]; [D356A] | | p.R545C | |
| **Perinatal history** | | | | | | | |
| Preterm/Term | Term | Term | Term | Preterm | Preterm | Preterm | Preterm |
| HC at birth cm (%ile) | 32 (5%ile) | 33 (10%ile) | 32.5 (5–10%ile) | 33 (10%ile) | 33 (10%ile) | unk | unk |
| Weight at birth kg | 3.2 | 2.5 | 3.0 | 2.2 | 2.3 | unk | unk |
| Length at birth (cm) | 47 cm | 49 cm | 48 cm | unk | unk | unk | unk |
| Complications | No | No | No | No | Hypoglycemia | No | No |
| **Psychomotor development** | | | | | | | |
| Gross motor skills | Severe delay | Severe delay | Severe delay | Mild-moderate delay | Mild-moderate delay | Severe delay | Moderate delay |
| Fine motor skills | Severe delay | Severe delay | Severe delay | Mild-moderate delay | Mild-moderate delay | Absent | Absent |
| Language | Absent | Absent | Absent | Absent | Absent | Delay | Delay |
| Social | Severe delay | Severe delay | Severe delay | mild-moderate delay | mild-moderate delay | Severe delay | Severe delay |
| Intellectual disability | Severe | Severe | Severe | Severe | Severe | Profound | Severe |
| Degenerative course | Y | Y | Y | Y | Y | Y | Y |
| **Neurological examination** | | | | | | | |
| Age at last exam | 9 yrs | 7 yrs | 6 yrs | 14 yrs | 7 yrs | 20 yrs | 17 yrs |
| HC at last exam (−SD) | 42 cm (−5 SD) | 45 cm (−4 SD) | 43.9 cm (−4 SD) | 49.2 cm (−4 SD) | 47.5 cm (−4 SD) | 46.5 cm (−4.2 SD) | unk |
| Tone | Low | Low | Low | Low | Low | Low | Low |
| **Seizures** | | | | | | | |
| Onset | 5 mos | 12 mos | 1 yrs | 1 yrs | 6 mos | unk | unk |
| Type | GTC | GTC | GTC | GTC | GTC | GTC | NA |
| **Brain MRI** | | | | | | | |
| Age of MRI | 7 yrs | 5 yrs | 1 yrs | 14 yrs | 7 yrs | 20 yrs | unk |
| Findings | Global atrophy | Global atrophy | Global atrophy | Global atrophy | Global atrophy | Global atrophy | unk |
| **Other** | | | | | | | |
| Dysmorphism | Maxillary hypoplasia, progeroid features, arachnodactyly | Progeroid appearance, arachnodactyly | Progeroid features, arachnodactyly | Arachnodactyly, cubitus valgus, pes planovalgus | Arachnodactyly, cubitus valgus, pes planovalgus | Contractures | Scoliosis |
| Renal | Normal | Normal | Normal | Medullary sponge kidney | Normal | Normal | Normal |
| Hepatic | Normal | Normal | Normal | Hepatic hemangioma | Normal | Normal | Normal |
| Cataracts | Normal | Normal | Normal | Normal | Normal | Yes | Yes |

+ positive for consanguinity, HC head circumference, SD standard deviations, yrs years, unk unknown, mos months, Y yes, GTC Generalized Tonic-Clonic seizure.

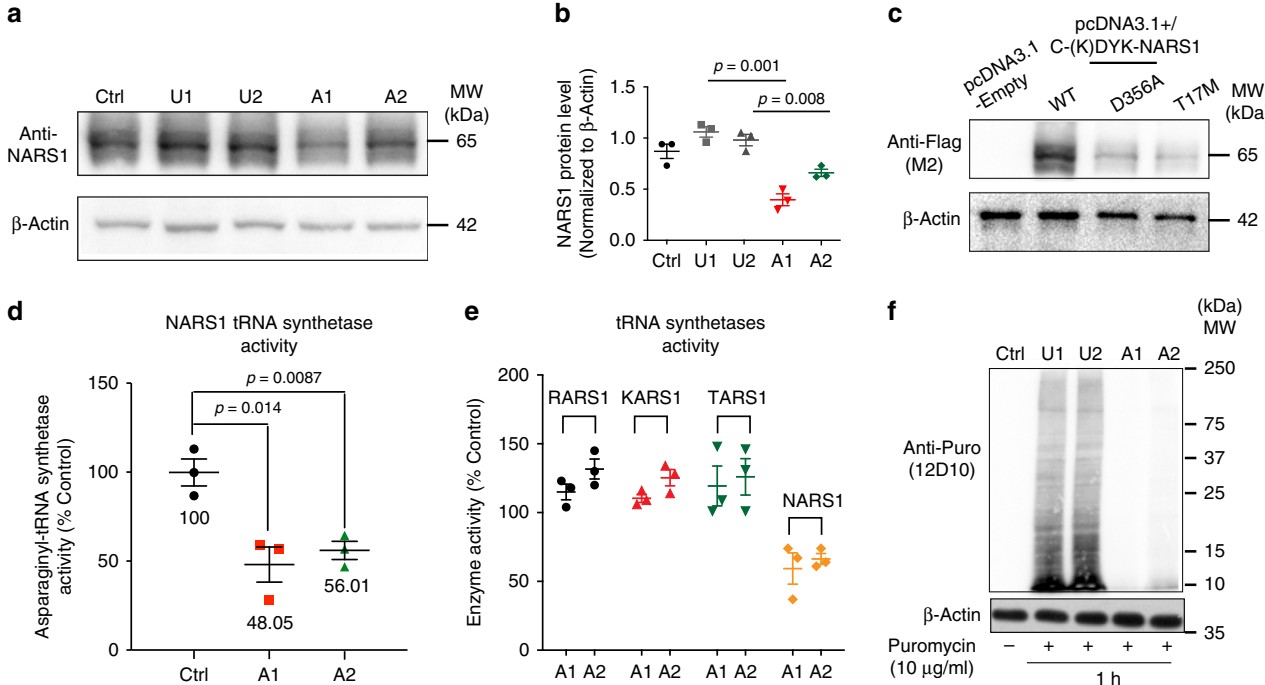

**Fig. 2 Patients cells display loss of *NARS1* functions. a** Fibroblast cells from two unaffected (U1 and U2) and two affected (A1 and A2) show reduced levels compared with actin control. **b** Quantification results for WB, $n = 3$, Error bar: ± SD. **c** Overexpression in cDNA expressing patient variants in HEK293T cells showed unstable NARS1 protein. **d** Reduced NARS1 tRNA synthetase activity in patient fibroblasts. $n = 3$, Error bar: ± SD. **e** Other tRNA-ARS activity not impaired in *NARS1* mutated cells. TARS: Threonyl-tRNA synthetase; KARS: lysyl-tRNA synthetase; RARS: arginyl-tRNA synthetase. $n = 3$, Error bar: ± SD. **f** Patient IPSCs showed reduced puromycin incorporation after 1 h pulse, indicating reduced protein synthesis. For the quantification data in (**b**) and (**d**), $n = 3$ represents three independent biological replicates, two-tailed $p$ values based upon Student $t$-test with Holm–Šídák multiple comparison correction, and individual $p$ value was shown in (**b**, **d**); Source data are provided as a Source Data file.

(Supplementary Fig. 3a, b, Supplementary Data 1). iPSCs from affected individuals were generally indistinguishable in measures of pluripotency and cellular health (Supplementary Fig. 3c, d)[43]. *NARS1* mRNA was expressed during NPC generation, strongest at day in vitro (div) 10–28, then dropped after div 35 (Supplementary Fig. 3e, f). 1 h EdU pulse showed significantly reduced DNA synthesis, but small pockets of proliferative cells remained in the NPCs cultures from affected patients (Supplementary Fig. 3g–i).

To model microcephalic phenotypes, we next generated 3D cultures of cortical brain organoids (COs) according to standard protocols (Fig. 3a, see methods)[44–48]. CO diameter was assessed as a function of days in vitro (divs). At div 28, organoid diameters were indistinguishable in the four individuals. However, by div 52, affected organoids were noticeably smaller than unaffected organoids (Fig. 3b, c, orange bar). CO sizes were binned by diameter as <1 mm, 1–5 mm, and > 5 mm[47,49]. By div 90 the vast majority of unaffected COs were > 5 mm, whereas none of the affected COs were > 5 mm, and in fact most were <1 mm diameter. Daily monitoring suggested that these differences did not represent shrinkage, as affected COs rarely achieved size > 1 mm. These results suggest that COs from *NARS1* patients are reduced in size, modeling patient microcephaly.

**Patient COs show reduced generation of RGCs**. In COs, SOX2+-RGCs are typically found in rosettes, surrounded by TUJ1+ post-mitotic neurons[44,50]. While COs from unaffected individuals showed well-formed compacted SOX2+ rosettes surrounded by TUJ1+ post-mitotic neurons, COs from affected individuals were poorly organized, lacked well-formed rosettes or representation of TUJ1+ cells. Although SOX2+ rosettes did not take on typical spherical appearances, there were

irregularly-shaped cavities that had faintly SOX2+ cells surrounding, but few strongly positive SOX2+ cells (Fig. 3d, e). The number of typical spherical SOX2+ rosettes, as well as the thickness of SOX2+ rosettes, which were measured as the diameter, were significantly decreased in affected COs compared to unaffected COs (Fig. 3f, g). The length of the strands of acetylated alpha Tubulin staining (c-Tub), a marker for mitotic spindles[51], was additionally used to measure the diameter (i.e., thickness) of SOX2+ rosettes, and showed a significant decrease in affected COs compared to the unaffected COs (Supplementary Fig. 4a–c). The mRNA expression level for the RGC marker PAX6 and the mature neuronal marker MAP2 were also significantly reduced in affected COs compared with controls, showing reduced immunostaining intensity. These data suggest failed RGCs generation, with subsequent depletion of post-mitotic neurons (Supplementary Fig. 4d, e).

**Single cell-RNA seq reveals a cell cycle defect in RGCs**. We next performed sc-RNA-seq to dissect the cellular composition in unaffected and affected COs (Fig. 4a). We first aggregated sc-RNA-seq reads from both unaffected and affected COs, and analyzed ~10,000 single cells at div 52. We identified four major cell clusters (RGC, Neuron, IPC, Astrocyte) and 23 sub-clusters, based on the expression of known marker genes (Fig. 4b–d, Supplementary Figs. 5c, 6 and 7a, Supplementary Data 2 and 3)[52–54].

Within the RGCs population, we found that several sub-clusters were dramatically reduced in affected COs (Fig. 4e), identifying genes involved in patterning and cell cycle, such as *MESIS*, *NAP1L1*, as well *as CCND2*, a regulator of the G1-S transition (Supplementary Data 3). GO-term gene functional analysis confirmed this result, which implicated a role for *NARS1* in cell cycle control (Fig. 4f). qPCR validation with cDNA from

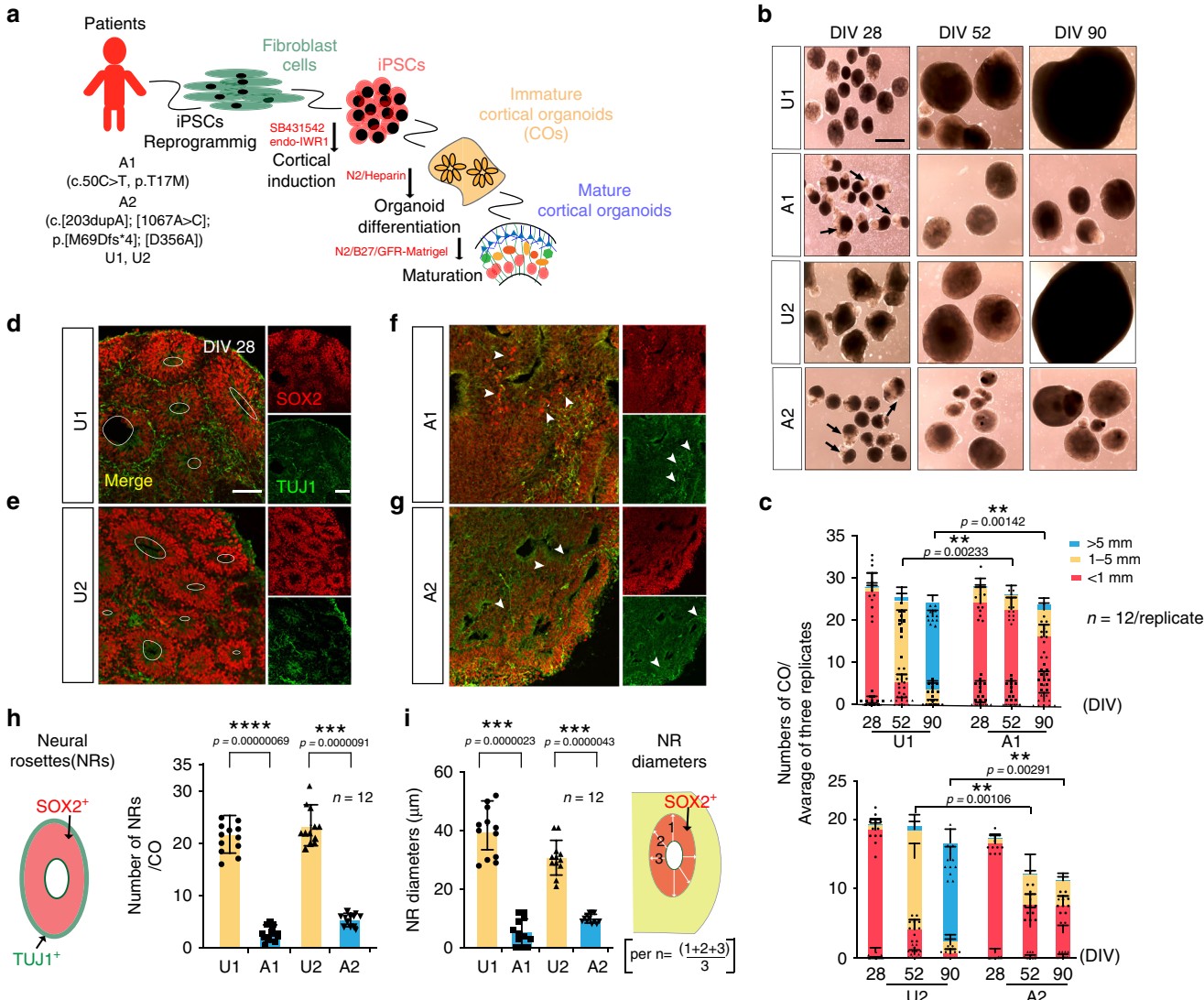

**Fig. 3 NARS1 patient COs show reduced size and impaired differentiation. a** Schematic of experimental work flow. **b** COs generated from iPSCs A1 and A2 showed reduced size compared with controls at DIV28, 52 and 90. Scale bar: 400 μm. **c** COs binned by diameter show impaired size in patients compared with controls. n = 12 (n represents 3 independent biological replicates generated from iPSCs at different passages (<25), and four times technical replicates for each of the independent biological replicate), Student t-test with Holm–Šídák multiple comparison correction was used to determine the two-tailed p value, **p < 0.01. Error bar: ± SD; Data are presented as mean values ± SD. The exact p value was shown in Fig. 3c. **d–g** Impaired morphology and cellular expression in patient COs, based upon SOX2 and TUJ1 expression. Unaffected COs shown in (**d**) (U1), (**e**) (U2), affected shown in (**f**) (A1), (**g**) (A2). Dashed circle represents inner circumference of neural rosette. White arrowheads: small clusters of SOX2+ cells in mutant COs. Scale bar = 100 μm; (**h**) Schematic and quantification for neural rosette numbers in (**d–g**). Neural rosettes were counted if they had a SOX2+ inner ring surrounded by a TUJ1+ outer ring. **i** Quantification and schematic of neural rosettes diameters of (**d–g**). Neural rosette diameters were measured at three different angles, set apart by 45° each, then averaged. For (**h**) and (**i**), n = 12 (n represents 3 independent biological replicates generated from iPSCs at different passages (<25), and four times technical replicates for each of the independent biological replicate), Student t-test with Holm–Šídák multiple comparison correction was used to determine the two-tailed p value, ****p < 0.000001, *** < 0.00001. Error bar: ± SD. Data are presented as mean values ± SD. Source data are provided as a Source Data file.

unaffected COs and affected COs at several differentiation stages, confirmed reduced *CCND2* mRNA expression (Fig. 4g). Furthermore, pseudo-time analysis showed altered cell fate towards astrocytes at the expense of neurons in affected COs (Supplementary Fig. 7b–e).

**Reduced RGC viability in patient-derived COs**. To validate sc-RNA-seq results, we assessed CO neural rosettes by immunofluorescence for markers of proliferation. We observed reduced populations of Ki67+ cells, a cell-cycle marker, in rosettes from

affected compared with unaffected (Fig. 5a, b, Supplementary Fig. 8b, c). WB analysis confirmed a reduction in total Ki67 protein, also known as MKI67, in COs from the affected individual in family MIC-2116 (Fig. 5c). qPCR analysis with COs at different time points confirmed that reduced KI67 mRNA levels were evident as early as div18, although there were differences observed in individual patient lines (Supplementary Fig. 8a). Cleaved Caspase 3+ (CC3) cells were apparent in affected COs but barely detected in unaffected COs (Fig. 5a, Supplementary Fig. 8b, c). Staining for the M-phase marker pH3 supported a mitotic defect in affected COs (Fig. 5d, e). The apical polarity

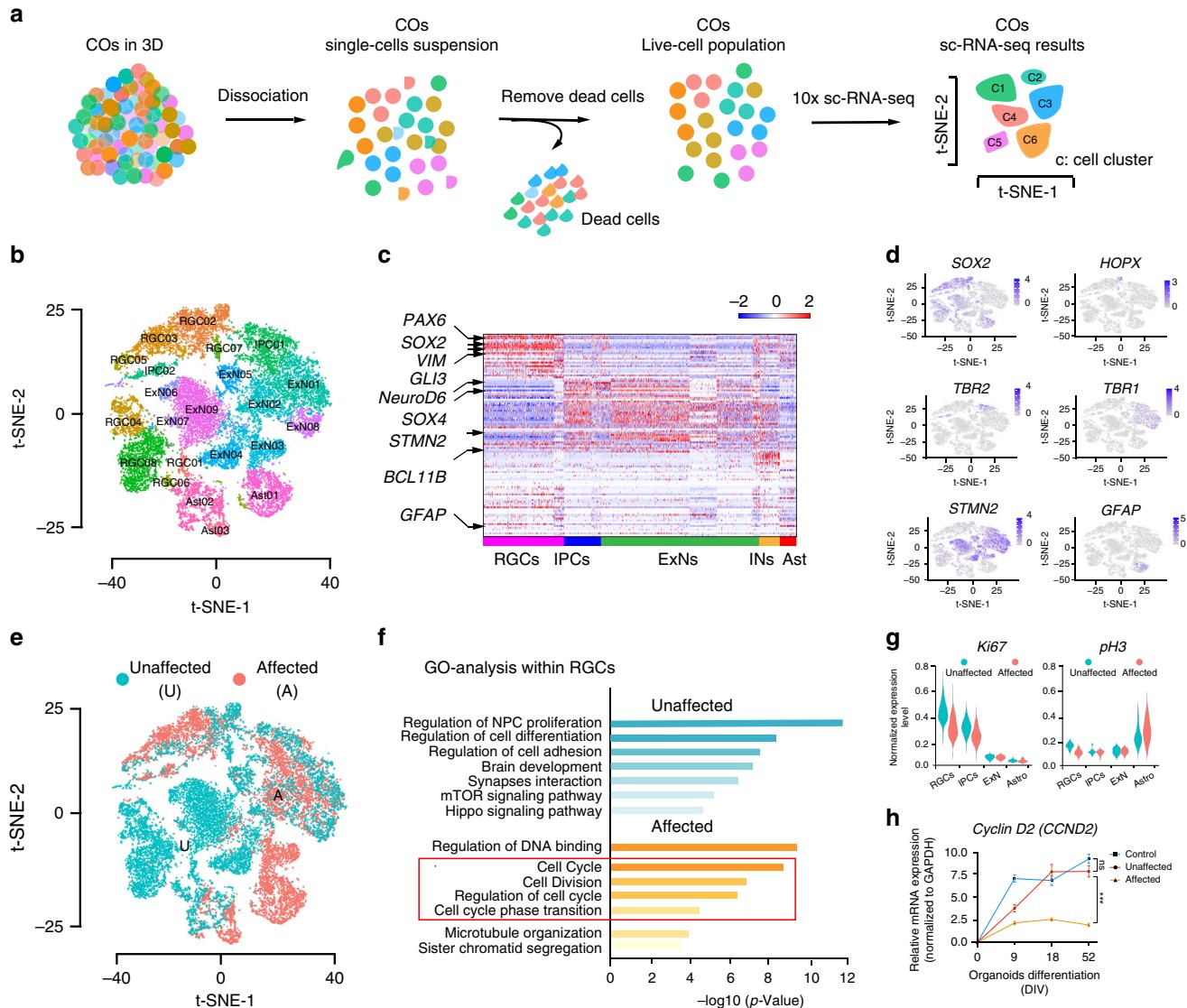

**Fig. 4 sc-RNA-seq suggests cell cycle defects in radial glial cells from affected individuals. a** Experimental work flow for sc-RNA-seq. **b** t-SNE-plot showed four major cell clusters and 23 detailed cell clusters in COs at div 52; RGC: Radial Glia Cells (with 8 sub clusters); Neurons, including both ExN: Excitatory Neuron (with 9 sub clusters) and INs: Inhibitory Neurons; IPC: Intermediate Progenitor Cells (with 2 sub clusters); Ast: astrocytes (with 3 sub clusters). **c** Heatmap for the four major cell clusters at div 52. Expression of *PAX6*, *SOX2*, *VIM* and *GLI3* highlighted to mark RGCs; *NEUROD6* and *SOX4* mark IPCs; *STMN2* and *BCL11B* mark ExNs and *GFAP* marks Asts. RPKM was used to calculate expression. **d** t-SNE plot shows the expression of marker genes in different cell clusters. *SOX2* marks RGCs; *HOPX* marks oRGs; *TBR2* marks IPCs; *STMN2* and *TBR2* mark ExNs; *GFAP* marks Ast. **e** t-SNE plot cell cluster distribution. **f** GO-functional enrichment analysis with differential expression genes (DEGs) highlights cell cycle related pathway in affected COs Log transformed *p* value was used to measure significance, the significance cutoff was set for $p < 0.05$. Unaffected: U1 + U2; Affected: A1 + A2 **g** Violin-plot represents the expression of *KI67* and *pH3* expression in the major cell clusters of unaffected and affected samples. **h** RT-qPCR shows a reduction of Cyclin D2 (*CCND2*) expression in affected COs. COs generated from affected and unaffected iPSCs harvested at div 9, 18 and 52 for RNA, following qPCR; Levels of *CCND2* was measured. *GAPDH* was used as control; Relative expression was normalized to cDN A derived from an unrelated control. $n = 12$ (*n* represents 3 independent biological replicates generated from iPSCs at different passages (<25), and four times technical replicates for each of the independent biological replicate), **$p < 0.01$. Error bars: ± SD. Data are presented as mean values ± SD. Source data are provided as a Source Data file.

marker αPKC was localized to apical junctions but the zones were often collapsed and poorly organized in affected COs (Fig. 5f).

COs displayed 3 cortical layers: ventricular like zone (VZ), sub-ventricular like zone (SVZ) and cortical plate-like zone (CP) (Fig. 5g), as reported[44], but affected COs showed the layers were not well organized. Distribution of TBR2, which represents intermediate progenitor cells, were not significantly altered within CO populations (Fig, 5h-i, Supplementary Fig. 8d)[55], but an obvious reduction was observed in the distribution of CTIP2+ cells in affected COs indicating loss of early born deep layer cortical progenitors (Fig. 5h, j, Supplementary Fig. 8d, e)[56].

Together, these results suggest loss of proliferation within progenitors in the affected COs correlated with morphological defects and smaller organoids.

## Discussion

Our study identified seven patients from three independent families with recessive microcephaly harboring four pathogenic variants in *NARS1*, validating *NARS1* loss as a cause for human microcephaly. All mutations disrupted highly conserved amino acid residues in critical regions of the protein, or led to premature stop codons. We found that patient-derived cells showed reduced

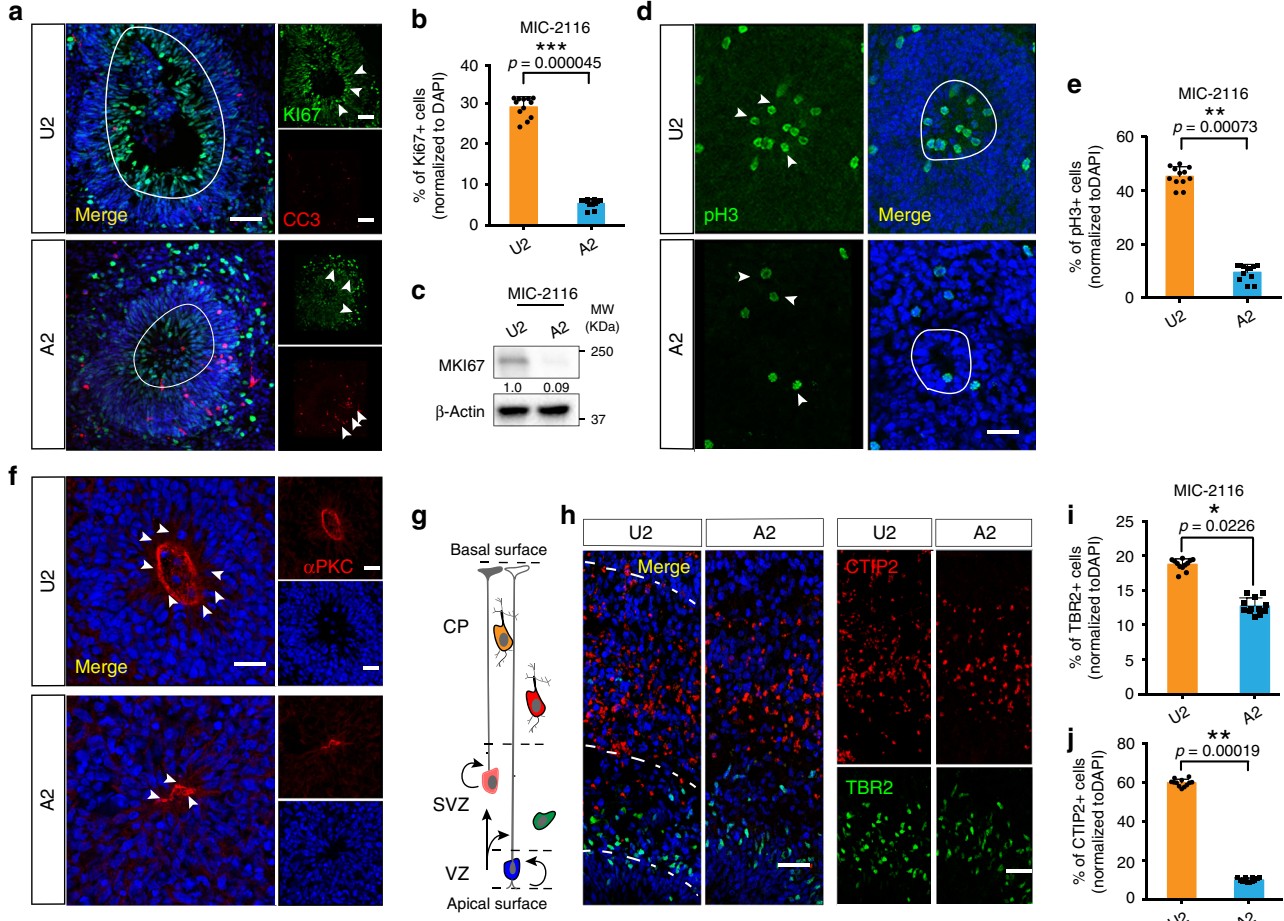

**Fig. 5 Reduced proliferation in affected cortical organoid radial glial cells. a** Reduced VZ-Ki67+ and increased CC3+ in affected COs at div 52. VZ regions: circle. Ki67+ and CC3+ cells; arrows. Scale bar = 100 µm. **b** Ki67+ cells normalized to DAPI+ cells; n = 12. **c** Reduced Ki67 levels in COs from affected at div 52; β-Actin loading control. **d** Reduced phospho H3+ (pH3+) cells in affected COs at div 52; VZ regions: circle, pH3+ cells: arrows. **e** Quantification of pH3+ cells normalized to DAPI+ cells; n = 12. **f** Reduced αPKC+ apical signal affected COs at div 52 (arrows). **g** Schematic for layer formation in COs. **h** COs from affected show less CTIP2+ deep layer neurons. COs at div 52. TBR2 (green): intermediate progenitor cells (IPCs) marked sub-ventricular zone (SVZ) in CO; CTIP2 (red): deep layer neurons marked cortical plate (CP) zone. **i, j** Quantification of TBR2 and CTIP2; n = 12. Scale bar = 100 µm. For (**b**, **e**, **i** and **j**), n represents 3 independent biological replicates generated from iPSCs at different passages (<25), and four times technical replicates for each of the independent biological replicate. Error bar: ± SD in all statistical results. For all statistical, Student t-test with Holm–Šídák multiple comparison correction was used to determine the two-tailed p value, *p < 0.05, **p < 0.01, ***p < 0.001. The exact p values for (**b**) is 0.00045, for (**e**) is 0.00073, for (**i**) is 0.0226 and for (**j**) is 0.00019. All data are presented as mean values ±SD. Source data are provided as a Source Data file.

cytoplasmic NARS1 transferase activity, and that patient-derived organoids modeled microcephaly, revealing defects in RGC proliferation.

The homozygous p.T17M variant located in the appended UNE-N domain resulted in reduced protein expression, and correlated with reduced aminoacylation activity in patient cells. The UNE-N domain is mammalian-specific and is not thought to play a direct role in tRNA transferase activity. Whether the UNE-N domain functions in transferase activity through regulation of super-molecular tRNA transferase scaffolds as proposed[7,9,10] was not addressed here. Enzyme activity of NARS1 in patient cells was significantly reduced, but not as dramatically as in other ARS mutant cell lines[3,4]. This may indicate that cells are exquisitely sensitive to reduced NARS1 activity, or that this activity is particularly important in brain function, where it is most strongly expressed (Supplementary Fig. 1c).

By taking the advantages of iPSC-derived COs in modeling the complexity of human brain development, we were able to recapitulate core features of microcephaly[44,56–58]. We found a significant reduction in organoid size during differentiation and

development, linked to a defect in neural rosette structure, showing reduction in numbers of neural progenitor cells in affected COs at early divs (div18). This observation was validated in a 2D neural differentiation system, and appeared to be a direct result of reduced cell division and increased cell death.

Combining cortical brain organoids and single-cell RNA-seq, we further characterized mechanism of microcephaly in COs[53]. We found reduced RGC populations within the affected COs, evidenced by reduced dimensions of neural rosettes and subsequent reduction in numbers of post-mitotic neurons. We found altered cell fate determination, biased towards astrocytic instead of neuronal cell fates in affected COs. NARS1 mutations affected gene expression in pathways related to cell cycle (CCND2) and proliferation (KI67), likely contributing to microcephaly. However, the variability of organoids with sc-RNA-seq, which is a common challenge for organoid modeling, needs to be further addressed by a rescue experiment in affected COs.

Although we found reduced NARS1 protein levels, reduced asparagine transferase activity, and reduced protein synthetic function in cells from affected patients still undetermined is the

mechanism by which these deficits lead to altered gene transcription, reduced cell proliferation, and premature differentiation that we observed in affected COs. Previous studies into mechanisms of microcephaly have identified links between defective protein synthesis and activation of genotoxic stress pathways, which may lead to cell death[1,19,21], but how defects in protein synthetic function are sensed by cells in ways that reduce entry into the cell cycle and alter cell fate will require additional studies. Developing potential treatments for microcephaly will require further understanding of the genetic causes and mechanisms leading to reduced cell number in patients. Our study, combining human genetics, CO modeling and single-cell transcriptomics may be one way to advance a mechanistic understanding of the cell composition and cell fate aspects of microcephaly.

## Methods

**Human subjects**. Patients were enrolled at the University of California San Diego according to an IRB-approved protocol. All subjects provided signed informed consent for study. DNA samples from each member of each family was assessed by SNP genotyping. Whole-genome or whole-exome sequencing was performed on blood-derived DNA from two members of each family, and results analyzed by Homozygosity Mapper software [homozygositymapper.org], SEQR software (Broad Institute), Varvis software (Limbus Medical Technologies GmbH) to identify blocks of homozygosity and biallelic pathogenic mutations. All variants passing criteria for pathogenicity were considered according to threshold of allele fraction <0.1% in gnomAD and the Greater Middle Eastern Variome, and predicted "damaging" by MutationTaster. Dermal fibroblasts were generated from skin punch biopsies, then cultured in DMEM with 20% fetal bovine serum (FBS) and pen/strep until confluent, and maintained at low passage in mycoplasma-free conditions.

**Mutation modeling**. Location of patient mutations in NARS1 was modeled with Pymol, utilizing PDB codes "UNE-N (4ZYA), Canonical Domain (5XIX)".

**Cell culture**. HEK293T cells were purchased from ATCC under an approval MTA and were not further authenticated and grown at 37 °C in DMEM medium supplemented with 15% FBS and 1X NEAA (non-essential amino acid), 1X GlutaMax. Patient fibroblast cells were cultured in DMEM medium in addition with 15% FBS plus NEAA and GlutaMax. H1-ESCs were cultured in mTeSR (StemCell Technologies).

**Authentication of cell lines used**. We used HEK293T cells (sex typed as female) and H1 ESC (sex typed as male) obtained from ATCC (CRL-11268™) and WiCell (WAe001-A), which were not further authenticated; All the patient iPSCs were reprogrammed by CIRM with non-integrating reprogramming method[59]. All iPSC lines reprogrammed at Cellular Dynamics, Inc (Madison WI) and passed the QCs of chromosomal integrity (by SNP microarray), pluripotency (by analysis of gene expression by qPCR of 48 mRNAs)[43] and identity confirmation (by genotyping), and tested negative monthly for the presence of mycoplasma [https://www.ncbi.nlm.nih.gov/pubmed/28400443].

**Cortical brain organoids generation and culturing**. Patients specific iPS cells were maintained in mTeSR and passaged according to manufacturer's recommendations. For organoids generation, iPS cells at 12 passages were dissociated into single cells with Accutase. In total, around 9000 cells were then plated in each well of an ultra-low-attachment 96-well plate (Corning, CLS3474) in "Cortical differentiation medium" with 20 μM Rock inhibitor, 5 μM SB431542, 3 μM endo-IWR-1 for the first 4d[47,49,60], Then, keep culturing the organoids for additional 13d with cortical differentiation medium in addition of 5 μM SB431542, 3 μM endo-IWR-1, the medium was changed every 2d. On 18d, the immature organoids were transferred into ultra-low-attachment 6-well plate, and cultured for another 16d in "Organoids differentiation medium", and media was changed every 3–4d[49,60]. On 35d, media was changed to "Maturation medium" supplemented with 1% Matrix High Concentration (HC), Growth factor reduced (GFR) Matrigel (Corning, 354263), and medium replaced every 3–4d for additional 34d, HC-GFR Matrigel was added fresh at each change. On 70d, media was changed to "Long-term maintain medium"[49] supplemented with 2% HC-GFR-Matrigel, 50 ng/ml BDNF (R&D, 248-BD-025/CF) and 50 ng/ml GDNF (R&D, 212-GD-010/CF) for long term organoid maintenance.

**Human neural progenitor cells generation and neuron differentiation**. Neural progenitor cells (NPCs) were obtained from embryoid bodies (EBs) originating from H1 ES cells and patient iPSCs formed with mechanical dissociation of cell clusters and plated in suspension in differentiation medium (DMEM/F12, 20%KO-

SR, 1XN2, 1XB27, 1XNEAA, 1XGlutaMax, 5 μM Dorsomorphin, 5 μM SB431542, 20 μM Y27632) in low attachment 6-wells plate for 7d, the medium was changed every 2d; then EBs were plated onto poly-ornithine/laminin-coated plate in NBF medium (DMEM/F12, 1XN2, 1XB27, 1XNEAA, 20 ng/μl bFGF). Rosettes were collected and passaged with accutase after 5-7d of NBF culturing, and resultant NPCs were cultured on ploy-ornithine/laminin plates in NBF medium for weeks. For neuron differentiation, NPCs were then cultured in NB (DMEM/F12, 1XN2, 1XB27, Neural Basal medium, 1X NEAA, 1X GlutaMax) medium for 7d; after 7d, the NB medium were then changed into BrainPhys™ Neuronal Medium (STEMCELL Technology, 05790), for 7d, and the cells were continuing to be cultured in BrainPhys™ Neuronal Medium in addition with 10 ng/ml BDNF, 10 ng/ml GDNF and 10 ng/ml NT3 for another 14d, and the resultant neurons were harvested for qPCR and immunostaining experiments.

**DNA constructs**. The cDNA of human NARS1 was purchased from GenScript under an approval MTA. Mutations were introduced with site-directed mutagenesis and all cloning was performed with Gibson isothermal method, then sequence-verified.

**SNP genotyping assay and digital karyotype**. iPSCs used in the study were harvested for genomic DNA, and 500 ng of total genomic DNA for each sample was used to run SNP genotyping assay with Ilumina Core-Exome-24 Kit run (Infinium CoreExome-24 v1.3 Kit (48 Samples), 20024662). The sequencing results was used to get the karyotype for each cell lines. All the sequencing and analysis were done by IGM core at UCSD. And the analysis has been done according to Illumina guideline.
[https://www.illumina.com/Documents/products/technotes/technote_cytoanalysis.pdf]

**Cyrosectioning and immunofluorescence**. Organoids were fixed in 4% paraformaldehyde for 15 min at room temperature followed by washing with PBST (PBS with 0.25% Tween20) three times for 5 min; The fixed organoids were then allowed to sink in 30% sucrose overnight at 4 °C, then embedded in 15%/15% gelatin/sucrose solution and cryosectioned at 20 μm. Organoids sections were permeabilized in 0.25% Triton X-100 for 15 min at room temperature, then blocked with 10% BSA in PBS for 1 h at room temperature. After washing with PBST 3 times for 5 min, sections were incubated with primary antibodies in 5% BSA in PBS at the following dilutions: SOX2 (Santa Cruz Biotechnology, sc17319, 1:100), TUJ1 (Biolegend, 801202, 1:1000), Cleaved Caspase 3 (Cell Signaling Technology, 9661S, 1:500), Ki67 (BD-Biosciences, 550609, 1:1000), PAX6 (BioLegend, PRB-278P, 1:300), phospho-Histone3 (Ser10) (Cell Signaling Technology, 9701, 1:500), CTIP2 (Abcam, ab28448, 1:500), TBR2 (Abcam, EPR19012, 1:250), pH3 (Abcam, ab80612, 1:500), αPKC (Abcam, ab4124, 1:250), ac-Tub (Sigma, T7451, 1:250) overnight at 4 °C, then washed three times with PBST for 10 min, then incubated with secondary antibodies (Alexa Fluor™ 488 donkey anti-mouse IgG (H + L), 1915874, 1:1000; Alexa Fluor™ 594 donkey anti-rabbit IgG (H + L), 1890862, 1:1000, Alexa Fluor™ 594 donkey anti-chicken IgG (H + L),703585155, 1:1000, Alexa Fluor™ 488 donkey anti-rat IgG (H + L), 712546153, 1:1000) together with DAPI (ThermoFisher Scientific, D1306, 1:50000) for 2 h at room temperature, washed with PBST three times for 5 min, and mounted with the Dako, S3023. All the images were taken with ZEISS LSM880 Airyscan, with post-aquisition analysis done in ImageJ-6.

**Real-time quantitative PCR**. Total RNA samples were isolated from whole organoids using RNeasy Mini Kit. 1.5 μg of RNA was used to generated cDNA using the SuperScript III First-Strand Synthesis Kit. Real-time qPCR was performed using iTaq Universal SYBR® Green Supermix in the Bio-Rad Real-Time PCR System, with conditions: 95 °C for 5 min, 45 two-step cycles at 95 °C for 10 s and 60 °C for 30 s.

**Dissociation of organoids**. For early-stage organoids that were cultured <1 month, dissociation was performed by using Accutase. Organoids were washed once with PBS, then incubated with 1 ml Accutase in 37 °C incubator for 10 min, then mightily trituration 3~5 times, and another 1 ml Accutase added and incubated at 37 °C for another 10 min, then added 3 ml Neuronal Medium (1/2Neural Basal, 1/2DMEM/F12, 1XN2, 1XB27, 1XNEAA, 1XGlutaMax, 10 ng/ml BDNF, 10 ng/ml GDNF, 20 μM Y27632) to stop Accutase dissociation. Gentle trituration was performed to obtain single-cell suspension. Dissociation of organoids over 60 div was performed by dissociation using AccuMax (STEMCELL Technology), washed with PBS once, then incubated with 1 ml AccuMax in 37 °C incubated for 20 min, then 3 ml Neuronal Medium added followed by gentle trituration to obtain single cells in suspension.

**Library preparation for 10× single-cell RNA-sequencing**. Cortical organoids were collected at div 52, for 10X droplet generation, combined and dissociated into single cells. Organoids were selected based as representative of the overall population of organoids within the dish. At least 3 organoids were selected from each condition for quantification, to ensure that individual variation in organoids was minimized, EasySep™ Dead Cell Removal Kit (STEMCELL Technology) was used

to remove the dead cells, live cells were re-suspended in 0.025%BSA/PBS in addition with 10 μM Y27632 (Selleck) at the concentration of 100 cells/μL to generate cDNA libraries with the Single Cell 3' Reagent Kits V2, according to the manufacturer's instructions. Briefly, cells were partitioned into nanoliter-scale Gel Bead-In-Emulsions (GEMs) with the 10x Chromium Controller (10X Genomics), then cells flowed at a limiting dilution into a stream of Single Cell 3' Gel Beads. Upon cell lysis and dissolution of the Single Cell 3' Gel Bead within the droplet, primers containing an Illumina P7 and R2 sequence, a 14 bp 10XBarcode, a 10 bp randomer, and a poly-dT primer sequence were released and mixed with the cell lysates and bead-derived Master Mix. Barcoded, full-length cDNA from poly-adenylated mRNA was then generated in each individual bead, then individual droplets were broken and homogenized before the remaining non-cDNA components were removed with silane magnetic beads (Invitrogen). The libraries were then size-selected, and the R2, P5 and P7 sequences were added to each selected cDNA during end repair and adapter ligation. After Illumina bridge amplification of cDNA, each library was sequenced using the Illumina Hiseq4000 PE100bp (PE: pair end) in Rapid Run Mode.

**Data processing of single-cell RNA-seq from chromium system**. Fastq files were aligned by CellRanger® count function with default setting. GRCh38/hg38 v12 was used as the reference genome. Data from all runs were aggregated with "aggre" function to ensure comparable read depth across runs, combined output file of all runs were loaded into R as a Seurat object[61], log normalized and scaled with a scale factor of 10,000. Cells with <200 or > 2500 genes expressed (UMI count greater than 0) were removed according standard analysis principle of Seurat. The top 2000 variable genes (HVGs) were identified with Seurat FindVariableGenes, using "vst" as the method. We used PCA and t-distributed stochastic neighbor embedding (t-SNE)[62], as our main dimension reduction approach. PCA was performed with RunPCA function (Seurat) using HVGs. Following PCA, we conducted JACKSTRAW analysis with 100 iterations to identify statistically significant (p value < 0.01) principal components that were driving systematic variation. We used t-SNE to present data in two-dimensional coordinates, generated by RunTSNE function in Seurat. Significant PCs identified by JACKSTRAW analysis (first 20 PCs) were used as input. Perplexity was set to 30 (default). t-SNE plots were generated using R package ggplot2[63]. Clustering was done first by establishing a shared nearest neighbor and then conducting Luvain-Jaccard analysis on the resulted graph using FindClusters function from Seurat with default setting. DEX analyses were conducted using Seurat function FindAllMarkers. Briefly, we took one group of cells and compared it with the rest of the cells, using a Wilcoxon rank sum test. For any given comparison, we only considered genes that were expressed by at least 50% of cells in either population, with log fold-change greater than 0.69. Genes that exhibit p values under 0.01 were considered statistically significant after multiple testing corrections. All violin plots were generated using ggplot2, and for Y-axis, we calculated the normalized expression level of certain genes, by natural-log transformed the feature counts for each cell and divided by the total counts for that cell, then multiplied by the scale factor using log1p. t-SNE plots were generated using t-SNE plot function from R package Seurat. Except otherwise noted, all heatmaps were generated with R function heatmap.3.

**Western blotting**. Western blotting was performed with the following antibodies: anti-NARS1 (Novus Biologicals, NBP1-31896, 1:1000), anti-Puromycin (12D10) (Millipore Sigma, MABE343, 1:1000), anti- Ki67 (BD-Biosciences, 550609, 1:2000), anti- β-Actin (Santa Cruz Biotechnology, sc-393933, 1:1000). Whole-cell lysis was generated with TNE (Tris, NP40, and EDTA) lysis buffer in addition of protein inhibitors (Cocktail, Roche). Quantification of each band was performed using Bio-Rad analyzer, the statistical analysis was done in Prism6. Protein levels measured with Bio-Rad analysis software. All the uncropped results are available in Supplementary Fig. 9 and Source Data file.

**tRNA-synthetase activity assay and Puromycin labeling assay**. Cell lysates (cytosolic fraction, excluding mitochondria) were incubated in triplicate at 37 °C for 10 min in a reaction buffer (50 mM Tris-buffer (pH 7.5), 12 mM MgCl₂, 25 mM KCl, 1 mg/mL bovine serum albumin (BSA in PBS), 0.5 mM Spermine, 1 mM ATP, 0.2 mM yeast total tRNA, 1 mM Dithiothreitol, 0.3 mM [$^{15}N_2$]-asparagine, [$^{13}C_4$, $^{15}N$]-threonine, [$D_2$]-glycine, [$^{15}N_2$]-arginine and [$D_4$]-lysine)[3]. The reaction was terminated using 33.3%TCA. Ammonia was added to release the labeled amino acids from the tRNAs. [$^{13}C_2$, $^{15}N$]-glycine and [$^{13}C_6$]-arginine were added as internal standards and the labeled amino acids were quantified by LC-MS/MS. Intra-assay variation was <15%. Threonyl-tRNA synthetase (TARS1), arginyl-tRNA synthetase (RARS1), and lysyl-tRNA synthetase (KARS1) activity were simultaneously detected as control enzymes; For puromycin labeling assay, cells were treated with 10 μM of puromycin for 1 h before harvest for western blot using to anti-puro-12D10 as described[42].

**Blinding of datasets**. NARS1 tRNA synthesis activity was measured by a blinded observer. Numbers of organoids and cells were counted by a blinded observer. scRNA-seq data analysis was blinded to the analyzer.

**Inclusion and exclusion criteria of data**. We did not set prior inclusion or exclusion criteria for data.

**Quantification and statistical analysis**. Unless otherwise specified, data are presented as mean ± standard error of the mean (SD). Significance was assessed by the Student t-test with Holm–Šídák multiple comparison correction using Graph-pad Prism8 software. In organoids modeling experiments, n represents number of biological experimental replicates. Immunofluorescence analysis, quantitative RT-PCRs, western blots, and biochemical assays were performed at least in triplicate. p value < 0.05 after correction for multiple comparison testing was considered statistically significant for all datasets. No statistical methods were used to predetermine sample size. And for results shown in Figs. 2a, c, f, 3b, d, e and 5a, c, d, f, h and Supplementary Fig. 2a, c, 3c, d, g, h and 4a, e, three times each experiment was repeated independently with similar results.

**Reporting summary**. Further information on research design is available in the Nature Research Reporting Summary linked to this article.

## Data availability

The whole-exome sequencing data from individuals from families 2116, 1433 and 91 in this study have been deposited in the database of Genotypes and Phenotypes (dbGaP) under accession phs000288.v2.p2. Single-cell-RNA-seq data of COs were generated at Institute of Genome Medicine genomic core at UCSD. The raw data were submitted to National Center of Biotechnology Information under the accession number of PRJNA590197. All other data supporting the findings of this study are available from the corresponding author upon reasonable request. The source data underlying Figs. 2a–d, 3c, h, i, 4h and 5b, e, i, j and Supplementary Figs. 2, 3f, i, 4c, d and 8a, c, d, f are provided as Source Data files.

## Code availability

GATK best practices pipeline was used for SNP and INDEL variant identification [https://gatk.broadinstitute.org/hc/en-us]. Variants were annotated with in-house software1 based on ANNOVAR for annotation and homozygous variant priorization was done using custom Python scripts (available upon request with jogleeson@ucsd.edu) variant filtering; Identify blocks of homozygosity and biallelic pathogenic mutations were analyzed by Homozygosity Mapper software, SEQR software, Varvis software; Cell Ranger-3.0.2 and Seurat 3.1.2 were used for single-cell RNA-seq analysis.

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

## Acknowledgements

We thank the families involved in this study, Dr. Ashely Marsh for comments on manuscript, Dr. Yao Tong and Dr. Paul Schimmel for suggestions on protein synthesis analysis and Dr. Junhao Li for suggestions on sc-RNA-seq analysis. Lu Wang (L.W.) is a guarantee of the NARSAD Young Investigator supported by Brain Behavior Research Foundation (BBRF-28771). David Sievert (D.S.) received funding from the California Institute for Regenerative Medicine. This work was supported by NIH grants R01NS048453, R01NS052455, the Simons Foundation Autism Research Initiative, and the Howard Hughes Medical Institute (to J.G.G.). Rady Children's Institute for Genomics Medicine for sequencing and bioinformatic support. Broad Institute (U54HG003067 to Eric Lander and UM1HG008900 to Daniel MacArthur), the Yale Center for Mendelian Disorders (U54HG006504 to Murat Gunel), Center for Inherited Disease Research for genotyping and sequencing support, and California's Stem Cell Agency (CIRM-IT1-06611) for patient iPSC lines.

## Author contributions

L.W. conducted experiments and wrote the manuscript; Z.L. performed bioinformatics analysis; D.S. helped with cell culture; D.E.C.S. and M.I.M performed aminoacylation activity analysis under supervision by G.S.S; V.S., H.H. R.O.R., M.K. and A.D.A performed patient recruitments; D.C. and H.H. performed clinical genetic analysis and genetic analysis; Y.L.W. supported protein structure analysis; S.G. performed Western analysis. J.G.G. supervised the project. All the authors approved the manuscript.

## Competing interests

The authors declare no competing interests.
