## [Peer Review File · Nature Communications]

Peer Review Information

Manuscript title: Biallelic loss of human NARS leads to microcephaly and impairs proliferation of radial glial cells in cortical brain organoids

Corresponding author name(s): Prof Joseph Gleeson

Editorial notes:

Reviewer comments & decisions:

Reviewer comments, first version:

Reviewer #1 (Remarks to the Author):

Wang and colleagues describe a study that implicates NARS1 variants in a cohort of patients with a recessive neurodevelopmental disorder that includes microcephaly. The arguments presented for pathogenicity are convincing: the variants support a recessive mode of inheritance and were identified in three families with an overlapping phenotype; functional studies reveal that the variants affect the primary function of NARS1 (tRNA charging) consistent with other ARS related disease; and studies in patient-derived organoids reveal a cellular phenotype consistent with other forms of microcephaly. This informative study is quite impactful as it represents the first report of NARS1 variants in human inherited disease and therefore expands the locus and allelic heterogeneity of ARS related disease. Furthermore, the organoid studies for the first time show the importance of an ARS enzyme for radial glial cell proliferation. There are only a few minor concerns, which have been outlined to the authors.

1. The new nomenclature for ARS loci (which I am not a big fan of) is 'ARS1' for the cytoplasmic and bifunctional, and 'ARS2' for mitochondrial. So, throughout the manuscript the gene and protein should be referred to as 'NARS1'.
2. 'ARS' should not be italicized as this does not refer to a specific locus but a gene family.
3. The authors should clarify if other ARS loci have been associated with primary microcephaly and if those cases only had microcephaly or if there were other phenotypes. Similarly, the authors should elaborate on the phenotypes found in the presented patients. Did they have additional phenotypes found in patients with other recessive, ARS-associated disease (e.g., liver dysfunction, lung disease, hearing deficits, brittle hair or nails, etc.); it would be surprising if only the CNS is involved.
4. Where the authors mention microcephaly and QARS, KARS, VARS, and CARS variants, the primary references for each of these papers should be in the main part of the manuscript.

5. The modest reduction in tRNA charging in patient cells (~50%) is rather surprising based on other studies on ARS related disease and the fact that NARS1 has a pLI of 0 indicating that haploinsufficiency isn't likely to be a problem. The authors should discuss these results (and the limitations of the assay used to generate them considering the patient tissues affected) in more detail.

Reviewer #3 (Remarks to the Author):

In this manuscript, Wang et al. perform a study of the human asparaginyl-tRNA synthetase (NARS). They analyzed whole exome sequencing from over 5000 individuals and found three families with biallelic NARS mutations. These patients displayed microcephaly, developmental delay, and loss of brain function. The authors then measured the protein expression and biochemical activity of NARS in subject cells and found that cells from affected individuals had impaired expression or function. Then, the authors differentiated 2D neuronal progenitor cells and 3D cortical brain organoids from patient and control-derived induced stem cells. They claim defects in proliferation, and a severe size and architectural deficits in cortical organoids from patients. Lastly, they used single cell RNA sequencing (10X Genomics) to resolve transcriptomes from brain organoids after ~2 months in vitro, using relatively standard methods (Seurat) to call cell clusters and to broadly define cell types. Overall, this important work. However, there are several major concerns regarding the human cellular models of disease that need to be addressed.

One of the main issues is the quantification of data. Throughout the paper it is unclear what the number of differentiation experiments, number of organoids or clones and number of cells is. It is essential to plot and perform statistics across clones. In the scRNA-seq, the authors should explicitly plot the distributions of cell types & clusters by sample. This needs to be quantified. Data points should be shown everywhere instead of columns (e.g., Figure 3C). Similarly, more experimental details should be provided. For instance, how were organoids selected for each experiment (especially for single cell)? Were any cells eliminated during cell processing or in the analysis (e.g., cells expressing non-ectodermal markers, cells expressing stress markers, etc)?

The authors claim that SOX2+ rosettes surrounding TUJ1+ cells are poorly organized/irregular in the patient organoids. However, the data is not quantified, and it is unclear how the authors conclude that lack of well positioned SOX2 cells/rosettes result in RGC generation failure. Although it might be possible that SOX2 rosettes are not forming because of mutant NARS protein, it is also plausible that there is a defect preceding RGC generation. Ideally, additional markers for RGC should be used for quantification.

The biochemical function is mildly impaired, but the phenotype is extreme. At what point during differentiation of organoids does the biochemical impairment impacts differentiation? The authors show NARS protein from whole cell lysates from fibroblasts in patient lines but they should also confirm NARS is expressed, and decreased, in cortical organoids specifically. Also, there is quite a difference between A1 and A2 that doesn't seem to reach significance but visually look different (Figure 2a-b). What is accounting for this difference in NARS protein between the two patients?

The characterization of hiPSC is incomplete. The authors should run SNP-assays or array CGH to confirm copy number variation changes and set a threshold on how many total kb deletion or duplications results in exclusion criteria. This should be provided in the methods with a rationale, as well as which lines were excluded and why. Additionally, traditional hiPSC staining (OCT, NANOG, etc) should be done on cultured hiPSC lines.

In Supp. Figure 5a, the authors show that cleaved caspase 3 (CC3) was increased in patient lines. Could the authors comment on the presence of a decrease in K67 activity and an increase in CC3?

It may be beyond the scope of this paper, but what are some potential mechanisms by which NARS affects RGC?

Other issues:

- It is unclear what 'BD' means in 1e.
- Figure 2d - why is the control activity 150% of control?
- Figure 4 - in (f) it's not clear what the GO enrichment corresponds to. All the affected cells vs unaffected, differential expression? cluster specific differences? Please clarify the directionality and the experiment
- Figure 4 (c, g) have no Y axis legend/title

Author rebuttal, first version:

Responses to reviewers

Reviewer #1 (Remarks to the Author):

Wang and colleagues describe a study that implicates NARS1 variants in a cohort of patients with a recessive neurodevelopmental disorder that includes microcephaly. The arguments presented for pathogenicity are convincing: the variants support a recessive mode of inheritance and were identified in three families with an overlapping phenotype; functional studies reveal that the variants affect the primary function of NARS1 (tRNA charging) consistent with other ARS related disease; and studies in patient-derived organoids reveal a cellular phenotype consistent with other forms of microcephaly. This informative study is quite impactful as it represents the first report of NARS1 variants in human inherited disease and therefore expands the locus and allelic heterogeneity of ARS related disease. Furthermore, the organoid studies for the first time show the importance of an ARS enzyme for radial glial cell proliferation. There are only a few minor concerns, which have been outlined to the authors.

Response. We thank R1 for these positive comments.

1. *The new nomenclature for ARS loci (which I am not a big fan of) is 'ARS1' for the cytoplasmic and bifunctional, and 'ARS2' for mitochondrial. So, throughout the manuscript the gene and protein should be referred to as 'NARS1'.*

Response. Done

2. *'ARS' should not be italicized as this does not refer to a specific locus but a gene family.*

Response. We agree and have made these changes.

3. *The authors should clarify if other ARS loci have been associated with primary microcephaly and if those cases only had microcephaly or if there were other phenotypes. Similarly, the authors should elaborate on the phenotypes found in the presented patients. Did they have additional phenotypes found in patients with other recessive, ARS-associated disease (e.g., liver dysfunction, lung disease, hearing deficits, brittle hair or nails, etc.); it would be surprising if only the CNS is involved.*

Response. R1 asks a good question. We now document other ARS genes showing microcephaly. We have expanded our clinical table to clarify phenotypes across different organs, shown in Table 1; From the newly ascertained information, we documented that

patient 2116-3-1 has medullary sponge kidney and hepatic hemangioma, and patients 91-3-1 and 91-3-2 have cataracts. The rest of the patients report no phenotypes across different ARS-associated diseases classes, such as liver dysfunction, lung disease, hearing deficits, brittle hair or nails, cataracts, peripheral neuropathy, skeletal dysplasia, cardiomyopathy, renal failure, or anemia.

4. Where the authors mention microcephaly and QARS, KARS, VARS, and CARS variants, the primary references for each of these papers should be in the main part of the manuscript.

Response. The primary references for QARS (11), KARS (36), CARS (37) and VARS (3, 4) have been added in the main manuscript line 79.

5. The modest reduction in tRNA charging in patient cells (~50%) is rather surprising based on other studies on ARS related disease and the fact that NARS1 has a pLI of 0 indicating that haploinsufficiency isn't likely to be a problem. The authors should discuss these results (and the limitations of the assay used to generate them considering the patient tissues affected) in more detail.

Response. We thank R1 for this excellent question. We re-quantified the results, to benchmark against a 100% control activity shown in Fig 2D, and now show a more dramatic reduction in residual enzymatic activity in patient lines, in the range of 50% reduction in activity. This reduced activity correlates with severely reduced protein synthetic function base upon our puromycin incorporation assay. We agree that haploinsufficiency is likely not disease-causing, as NARS1 has a pLI of 0, with 37.2 pLoF variants out of 71,702 genomes. We consider that haploinsufficiency in healthy individuals may not be disease-causing due to compensatory upregulation of NARS1 activity or expression, such that these individuals have more than the expected 50% activity, but since we have no patient cell lines with null alleles, we cannot test this hypothesis. We now discuss this in the main text line 254-259.

Reviewer #2 No comments

Reviewer #3 (Remarks to the Author):

In this manuscript, Wang et al. perform a study of the human asparaginyl-tRNA synthetase (NARS). They analyzed whole exome sequencing from over 5000 individuals and found three families with biallelic NARS mutations. These patients displayed microcephaly, developmental delay, and loss of brain function. The authors then measured the protein expression and biochemical activity of NARS in subject cells and

found that cells from affected individuals had impaired expression or function. Then, the authors differentiated 2D neuronal progenitor cells and 3D cortical brain organoids from patient and control-derived induced stem cells. They claim defects in proliferation, and a severe size and architectural deficits in cortical organoids from patients. Lastly, they used single cell RNA sequencing (10X Genomics) to resolve transcriptomes from brain organoids after ~2 months in vitro, using relatively standard methods (Seurat) to call cell clusters and to broadly define cell types. Overall, this important work. However, there are several major concerns regarding the human cellular models of disease that need to be addressed.

Response. We thank R2 for carefully reading and all these critical questions. We have addressed all the concerns point by point below.

One of the main issues is the quantification of data. Throughout the paper it is unclear what the number of differentiation experiments, number of organoids or clones and number of cells is. It is essential to plot and perform statistics across clones.

Response. We apologize for not presenting this data. In the original manuscript, we listed the number of replicates for each experiment, particularly in Figs. 2-3, but in the revision, we specify the type of replicates, and specify experimental, technical, averages from n organoids or cells. See in main text line 838~841 for Fig 2, and line 856-858, 864-868 for Fig 3. Numbers of organoids are now listed in Supplementary Table 3 sheet Fig2, Fig3.

In the scRNA-seq, the authors should explicitly plot the distributions of cell types & clusters by sample. This needs to be quantified. Data points should be shown everywhere instead of columns (e.g., Figure 3C).

Response. We agree, and now plot the distributions of cell types and clusters by sample with quantification in the Supplementary Fig 5. In Fig. 3C we now plot with mean plus datapoints instead of mean.

Similarly, more experimental details should be provided. For instance, how were organoids selected for each experiment (especially for single cell)? Were any cells

eliminated during cell processing or in the analysis (e.g., cells expressing non-ectodermal markers, cells expressing stress markers, etc)?

Response . We thank R2 for this question. In the methods, we now state that organoids selected for study were representative of the overall population of organoids within the

dish. At least 3 organoids were selected from each condition for quantification, to ensure representation (see in main text line 520-522). For cells dissociated from organoids, no live cells were excluded during cell processing (clarified in main text, method line 522-527), nor during data analysis cells. Individual cells with less than 200 or more than 2500 genes expressed (UMI count greater than 0) were removed from single cell RNAseq (clarified in main text, method line 546-551) according to the standard Seurat protocol. We noted a few cells that expressed non-ectodermal markers or markers of cell stress, which were retained in the analysis to keep full data representation.

The authors claim that SOX2+ rosettes surrounding TUJ1+ cells are poorly organized/irregular in the patient organoids. However, the data is not quantified, and it is unclear how the authors conclude that lack of well positioned SOX2 cells/rosettes result in RGC generation failure. Although it might be possible that SOX2 rosettes are not forming because of mutant NARS protein, it is also plausible that there is a defect preceding RGC generation. Ideally, additional markers for RGC should be used for quantification.

Response. We thank R2. To address the issue of quantification of the organoid organization, we now quantify the number of rosettes per organoid show in Fig. 3f. The diameters of the neural rosettes (based on average diameter of SOX2+ zone and length of ac-Tub strand) are quantitated in Fig. 3f and Suppl Fig. 4c. PAX6 staining was added in Supplementary Fig 4e to support SOX2 staining as an additional marker for RGC identity. In addition, we also show quantitatively that cultured NPCs show low proliferation rate in affected than in control, using pulsed Edu labeling (Supplementary Fig. 3g-i).

The biochemical function is mildly impaired, but the phenotype is extreme. At what point during differentiation of organoids does the biochemical impairment impacts differentiation? The authors show NARS protein from whole cell lysates from fibroblasts in patient lines but they should also confirm NARS is expressed, and decreased, in cortical organoids specifically.

Response. Excellent question. We now present western blot analysis of D18 and D28 as well as cultured NPCs and demonstrate reduction in NARS protein level at all time

points (Supplementary Fig. 2a, b). We now also include a puromycin pulse experiment from NPCs generated from patient iPSCs to demonstrate that at all time-points tested, there is a biochemical impairment in the patient cells (Supplementary Fig. 2c). We show that even in the dissociated NPC stage, there is a reduction in cell proliferation, which

was not observed in patient iPSCs or fibroblasts (Supplementary Fig 3g-h). Therefore, this seems to be a neural progenitor-specific phenotype, evident at the earliest stages.

Also, there is quite a difference between A1 and A2 that doesn't seem to reach significance but visually look different (Figure 2a-b). What is accounting for this difference in NARS protein between the two patients?

Response. We agree A1 and A2 show different protein levels, but these are from two different families with different severities and different biallelic mutations (Fig. 1a), so it is not surprising that NARS protein levels differ between the two patients. Nevertheless, the degree of NARS protein enzymatic activity is similarly reduced in A1 and A2 (Fig. 2e). We conclude that the mutation in A1 (Family-1433) leads to greater reduction in protein level, whereas the mutation in A2 (Family-2116) shows less reduction in protein level, but the residual protein is probably impaired to a greater extent than in A1, but with just a limited number of mutations, it is difficult to be more specific.

The characterization of hiPSC is incomplete. The authors should run SNP-assays or array CGH to confirm copy number variation changes and set a threshold on how many total kb deletion or duplications results in exclusion criteria. This should be provided in the methods with a rationale, as well as which lines were excluded and why. Additionally, traditional hiPSC staining (OCT, NANOG, etc) should be done on cultured hiPSC lines.

Response. We now present SNP genotypes on all 4 hiPSCs lines, and document absence of CNVs, deletions or duplications (Supplementary Fig. 3a, b, Supplementary Table 1). Additionally, immunofluorescent staining of OCT4 and NANOG for all four hiPSC lines showed evidence of pluripotency (Supplementary Fig. 3d).

In Supp. Figure 5a, the authors show that cleaved caspase 3 (CC3) was increased in patient lines. Could the authors comment on the presence of a decrease in K67 activity and an increase in CC3?

Response. We thank R2 for this excellent question. The decrease of the numbers of Ki67+ cells was specific to proliferating RGC populations, while CC3+ cells were

distributed throughout the organoid section, suggesting widespread apoptosis (see Fig 5a, and Supplementary Fig. 8a). We note in the text that some Ki67+ cell in patient organoids were found in ectopic locations, suggesting generalized disorganization. Our conclusion is that the reduced cell division and increased apoptosis contributes to the

5

reduced organoid size. The reduced proliferation seems to be cell autonomous since it was observed in dissociated NPCs (Supplementary Fig. 3e, f).

It may be beyond the scope of this paper, but what are some potential mechanisms by which NARS affects RGC?

Response. We now mention some potential mechanisms in the discussion, see in main text 286-299. Literature suggests that reduce protein synthesis may lead to genotoxic stress and activation of apoptotic pathways.

Other issues:

• *It is unclear what 'BD' means in 1e.*

Response. 'BD' means Blood

• *Figure 2d - why is the control activity 150% of control?*

Response. We re-quantified the results, to benchmark against a 100% control activity shown in Fig 2d. Raw data was provided in Supplementary Table 3.

• *Figure 4 - in (f) it's not clear what the GO enrichment corresponds to. All the affected cells vs unaffected, differential expression? cluster specific differences? Please clarify the directionality and the experiment*

Response. GO enrichment analysis was done within the RGC population, now clarified in main text line 212-214;

• *Figure 4 (c, g) have no Y axis legend/title*

Response. Corrected. See in main text line 569-572.

Reviewer comments, second version:

Reviewer #1 (Remarks to the Author):

The authors have satisfactorily addressed each of my concerns.

Reviewer #3 (Remarks to the Author):

The authors have overall addressed my questions. One issue is the high variability across organoids as shown in the new Suppl Fig 5. The overall noise in hiPSC disease modeling studies (and lack of an isogenic line in this study) combined with this high variability in organoid derivation may pose challenges in interpreting the results. The authors should at least make this very clear in the text and propose ways to address them in future studies.

Fig. 2c has a typo in 'empty'

What is the meaning of the 'X' in the raw WB results Fig. 5c panel in Supp. Fig. 9?

Is the scale bar in Supp. Fig. 3d 40 mm as stated in the legend?

Author rebuttal, second version:

Reviewer #1 (Remarks to the Author):

The authors have satisfactorily addressed each of my concerns.

We thank Reviewer 1 for positive feedback.

Reviewer #3 (Remarks to the Author):

The authors have overall addressed my questions. One issue is the high variability across organoids as shown in the new Suppl Fig 5. The overall noise in hiPSC disease modeling studies (and lack of an isogenic line in this study) combined with this high variability in organoid derivation may pose challenges in interpreting the results. The authors should at least make this very clear in the text and propose ways to address them in future studies.

We thank Reviewer 3 for good suggestion and positive feedback. We have discussed the variability across organoids in sc-RNA-seq data and propose ways to better address this problem in the discussion part. And we have answered the questions point by point here,

Fig. 2c has a typo in 'empty'

We thank R3 for carefully reading. And we have fixed this issue in the revised manuscript.

What is the meaning of the 'X' in the raw WB results Fig. 5c panel in Supp. Fig. 9?

We thank R3 for this question. The "X" indicates the none-related band, which is not used in this manuscript.

Is the scale bar in Supp. Fig. 3d 40 mm as stated in the legend?

We thank R3 for this question and the scale bar in Supp. Fig. 3d is 400 μm .

Reviewer comments, third version:

Author rebuttal, third version: